# In situ real-time gravimetric and viscoelastic probing of surface films formation on lithium batteries electrodes

Vadim Dargel[1], Netanel Shpigel[1], Sergey Sigalov[1], Prasant Nayak[1], Mikhael D. Levi[1], Leonid Daikhin[2] & Doron Aurbach[1]

It is generally accepted that solid–electrolyte interphase formed on the surface of lithium-battery electrodes play a key role in controlling their cycling performance. Although a large variety of surface-sensitive spectroscopies and microscopies were used for their characterization, the focus was on surface species nature rather than on the mechanical properties of the surface films. Here we report a highly sensitive method of gravimetric and viscoelastic probing of the formation of surface films on composite $Li_4Ti_5O_{12}$ electrode coupled with lithium ions intercalation into this electrode. Electrochemical quartz-crystal microbalance with dissipation monitoring measurements were performed with LiTFSI, $LiPF_6$, and $LiPF_6 + 2\%$ vinylene carbonate solutions from which structural parameters of the surface films were returned by fitting to a multilayer viscoelastic model. Only a few fast cycles are required to qualify surface films on $Li_4Ti_5O_{12}$ anode improving in the sequence $LiPF_6 < LiPF_6 + 2\%$ vinylene carbonate $<<$ LiTFSI.

[1] Department of Chemistry and BINA (BIU Institute for Nano-Technology and Advanced Materials), Bar-Ilan University, Ramat Gan 5290002, Israel. [2] School of Chemistry, Raymond and Beverly Sackler Faculty of Exact Sciences, Tel Aviv University, Tel Aviv 6997801, Israel. Correspondence and requests for materials should be addressed to D.A. (email: aurbach@biu.ac.il)

The formation and growth of solid–electrolyte interphase (SEI) on the surfaces of metallic Li, Li-alloy-based anodes, and most of the cathodes for Li-ion batteries strongly affect their long-term stability and capacity retention[1–4]. SEI-type surface films are formed due to chemical/electrochemical reactions of the components of electrolyte solutions (such as solvent molecules, salts, and additives) on the electrode surface, appearing as thin solid films of mixed organic–inorganic nature. Playing the role of interphase between the electrode bulk and the electrolyte solution, the surface films must be good Li-ion conductors but electronic insulators, in order to prevent further side reactions and continuous growth of surface films. Otherwise, significant consumption of active Li-ions from the electrodes during cycling (i.e., capacity fading) and deterioration of the electrodes rate capability (via a decrease in ion conduction) takes place during their cycling. It is therefore important, from the fundamental and practical points of view, to develop continuous in situ probing of the electrical and mechanical "state-of-health" of the forming SEI in battery electrodes in contact with various electrolyte solutions under diverse cycling conditions.

We have chosen for this study highly important challenge: understanding of the surface response of $Li_4Ti_5O_{12}$ electrodes denoted as LTO. These electrodes which redox potential is relatively high (1.5 V vs. Li) are considered as very stable and fast anodes for Li-ion batteries, in applications requiring very prolonged cycling (e.g., load leveling)[3]. They have a very intriguing surface chemistry that affects their performance, which was not explored yet properly[4].

Key elements of the SEI-type surface chemistries on different battery electrodes have been carefully studied via a wide use of surface-sensitive spectroscopic and microscopic tools[5–13], non-destructive nuclear magnetic resonance[8], and advanced electro-analytical methods[14–17], especially, high-precision coulometry[18,19]. However, much less is known about their mechanical properties as individual phases (e.g., whether they are stiff or soft, porous or non-porous, experiencing large or small intercalation-induced volumetric changes) in real time during their nucleation and growth as thin solid films from electrolyte solution components.

The quartz-crystal microbalance with dissipation monitoring instrument (QCM-D, for electrochemical measurements abbreviated as EQCM-D)[20–22], a powerful commercial instrument of general analytical use, is especially suitable for quantitative characterization of the viscoelastic properties of polymeric coatings and various soft objects (such as vesicles, polyelectrolyte layers, viruses, and growth of biomasses (biofouling))[20,21,23–27]. Before introducing EQCM-D for the study of composite electrodes, multi-harmonic measurements were used successfully for studying the electrochemical and mechanical properties of electronically conducting polymers. These measurements are based on network analysis that deals with the admittance of quartz crystals with rigidly attached electrodes, working in their thickness-shear resonance mode[28–32]. Gravimetric characterization of Li-battery electrodes[12,33–35] was performed, in most cases, by using the sole fundamental frequency rather than monitoring the accompanying resonance width change[36–39]. EQCM-D was recently used for a detailed study of the properties of surface films formed on Sn metallic films and a composite electrode during their lithiation in aprotic solutions[7,9,13,40].

EQCM-D methodology has the ability to monitor gravimetric and viscoelastic changes in composite battery electrodes caused by the formation of SEI coupled with Li-ion intercalation into battery electrodes. Acoustic multilayer viscoelastic modeling is used to extract the structural characteristics of the electrodes from the related EQCM-D data. A high-voltage $Li_4Ti_5O_{12}$ (LTO) spinel anode was chosen in order to clarify the controversial views

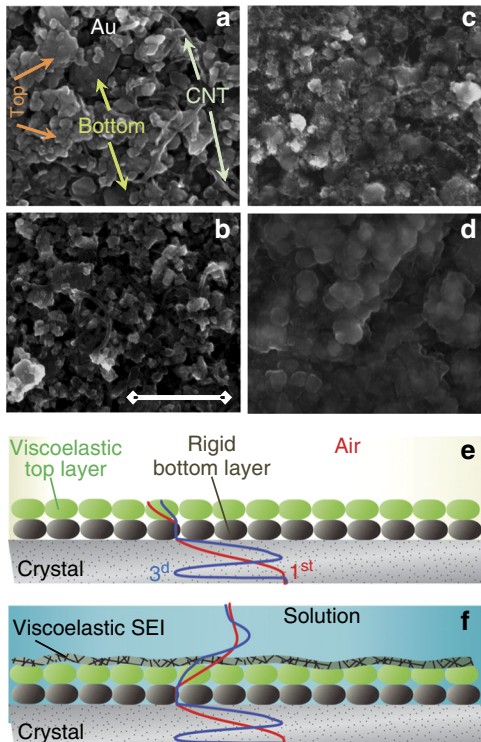

**Fig. 1** SEM images of LTO electrodes on quartz-crystal surface and their model presentation. **a** Pristine electrode, **b** cycled in EC-DMC 1 M LiTFSI, **c** 1 M LiPF$_6$, and **d** 1 M LiPF$_6$ + 2% VC solutions (scale bar = 500 nm). **e** Sketch of velocity profiles for rigid and viscoelastic layers of LTO electrodes in air and **f** in electrolyte solution. Velocity profile of shear wave crossing the multilayer assembly is exemplified for fundamental frequency and third overtone order

concerning the existence of SEI on the LTO electrodes' surface[5–7,10,11]. LTO is a zero-stress material that facilitates the assessment of gravimetric and viscoelastic properties of the SEI on its surface.

We report herein that different Li-battery electrolyte solutions can be effectively and quickly screened in research EQCM-D cells, facilitating the selection of optimal solution compositions. In order to demonstrate convincing models and methodology, here we select three electrolyte solutions: EC + DMC (ethylene carbonate + dimethyl carbonate) with LiTFSI (lithium bis(trifluoromethane sulfonyl) imide), LiPF$_6$ and LiPF$_6$ + 2% VC (vinylene carbonate). The quality of SEI on $Li_4Ti_5O_{12}$ electrode (easily recognizable from the short-time EQCM-D data) improves in the sequence LiPF$_6$ < LiPF$_6$ + 2% vinylene carbonate << LiTFSI. A similar trend can be obtained from the coin-cell experiments, however, after many hundreds of cycles. The interpretation of the electrolyte effect on the intrinsic mechanical properties of the SEI and the electrodes' capacity retention requires a careful consideration of the "floodness" factor, $S_{flood}$, defined as the ratio of the mass of the solution to the mass of the immersed electrode.

## Results

**Viscoelastic modeling of thin composite electrode coatings**. We used viscoelastic Voigt-type models as part of the acoustic multilayer formalism, which was previously developed for polymeric coatings, layers of vesicles, and lipid bilayers probed by QCM-D at different overtone orders (see "Methods" section)[25,41–44]. However, the model must be adapted to the composite LTO

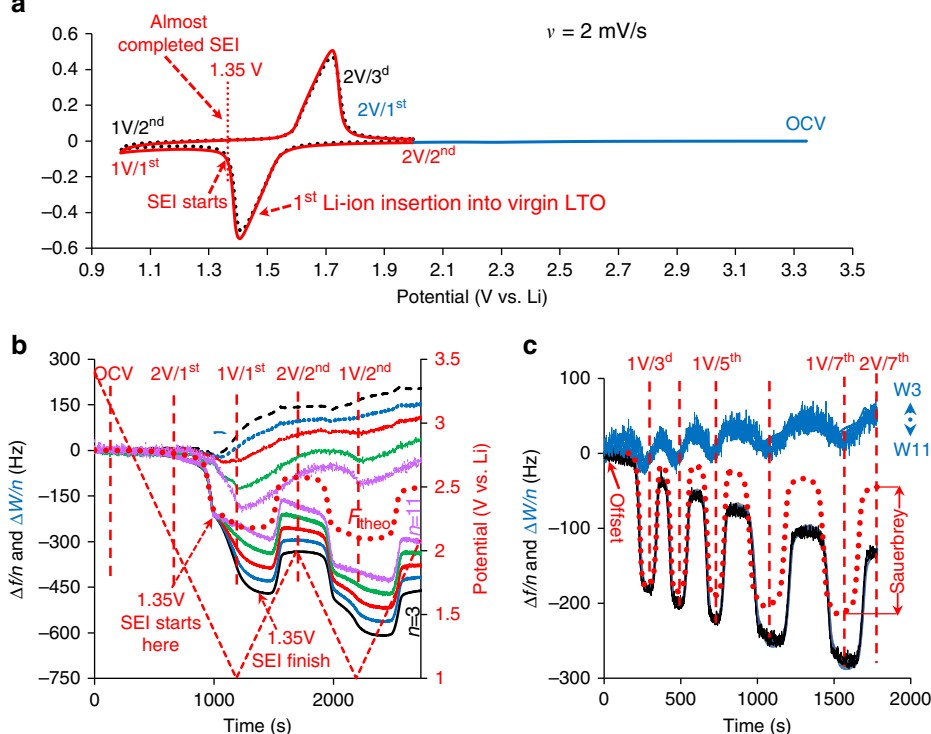

**Fig. 2** Characterization of LTO electrodes in EC-DMC 1 M LiTFSI solutions. **a** Two first sequential CV cycles from OCV to 1 V and then to 2 V, as indicated. The formation of SEI starts at 1.35 V during the first discharge and is almost completed at 1.35 V during the subsequent charge. **b**, **c** Frequency and resonance width changes at different overtones, *n* (shown by different colors) as functions of time during the first two initial cycles and during the subsequent five cycles, respectively. The dotted red line ($F_{theo}$) denotes the frequency shift calculated from the intercalation charge using the Faraday law and the Sauerbrey equation. The dashed red lines denote the potentials at which viscoelastic analysis has been carried out. In **b** W3↔W11 denotes the range of the resonance width changes for odd harmonics from *n* = 3 to 11

electrode ensuring complete EQCM-D characterization of this electrode in: (i) gas/air phase, (ii) in solutions under open-circuit voltage (OCV), and (iii) in solutions under applied potential. SEM images of these electrodes in the pristine state (Fig. 1a) and after cycling in the three selected solutions (Fig. 1b–d) are required in order to formulate an adequate viscoelastic model. The images shown in Fig. 1b–d clearly indicate the formation of surface films of different morphologies and thickness (their composition was further studied by XPS (X-ray photoelectron spectroscopy), as discussed below). The increase in the thickness of the surface films is inferred from the progressive smearing out of the separations between the individual electrode nanoparticles. The properties of the composite electrodes can be found by probing their viscoelasticity in the gas/air phase, according to the approach developed earlier[6]. A detailed description can be found in the "Methods" section.

In brief, multi-harmonic EQCM-D has two output characteristics, namely, resonant frequency, *f*, normalized by overtone order (*n*), *f*/*n*, and the dissipation factor, *D*, defined as the ratio between the full resonance peak width, *W*, and the resonance frequency, *f*: $D = (W/n)/(f/n)$. When the electrode coating is rigidly attached to the crystal surface, it moves in synchrony with the surface during crystal oscillations ("no-slip" condition). Alternatively, the viscoelastic coating changes the velocity profile across its width. This is shown schematically in Fig. 1e for a LTO electrode in the gas phase, and in Fig. 1f for the same electrode in contact with solution. Thus, the wave of sound penetrating a multilayer assembly comprised of stiff or viscoelastic layers affects the experimental $\Delta D$ and $\Delta f/n$ changes, whereas their viscoelastic parameters are selectively calculated during a fitting procedure. When measured in the air, LTO electrodes show both

*n*-dependent frequency shift, $\Delta f/n$, and an *n*-dependent change of $\Delta D$. Analysis of these shifts using a Voigt-type model reveals that the electrode layer consists of two acoustic layers, one is a rigid bottom layer and the other is a viscoelastic top layer (the viscoelastic analysis is shown in Supplementary Fig. 1 and graphically sketched in Fig. 1e, f). The viscoelastic model is used for acoustic characterization of the particulate composite LTO electrodes layers relying on the effective-medium theory[45]. In reality, a few rather than a single feature of the composite electrode structure affect the $\Delta f/n$ and $\Delta D$ changes. For this reason, we have compromised the loading mass density of the electrode coatings in the range between 60 and 90 μg/cm² (larger mass densities make multi-harmonic measurements unreliable). Polyvinylidene fluoride (PVdF) served as the binder, and LTO electrode was chosen because of its negligibly small intercalation-induced volume change. This choice allows us to fix the viscoelastic parameters of the electrode reducing the number of factors affecting $\Delta f/n$ and $\Delta D$ changes to the single one, namely, due to the formation and growth of SEI on the top of the electrode layer. More details on the important features of the application of viscoelastic model to particulate composite materials are given in "Methods" section.

**SEI formation on LTO electrodes in EC + DMC/LiTFSI solutions.** Figure 2a shows the initial potentiodynamic scan from OCV to 2 V vs. Li and the subsequent two cycles between 2 and 1 V. Figure 2b presents the related $\Delta W/n$ and $\Delta f/n$ changes (offset at *t* = 0 relates to the OCV), whereas Fig. 2c corresponds to the subsequent cycling with $\Delta W/n$ and $\Delta f/n$ offset at 2 V for the third cycle. The dashed red lines denote the potentials for which $\Delta W/n$

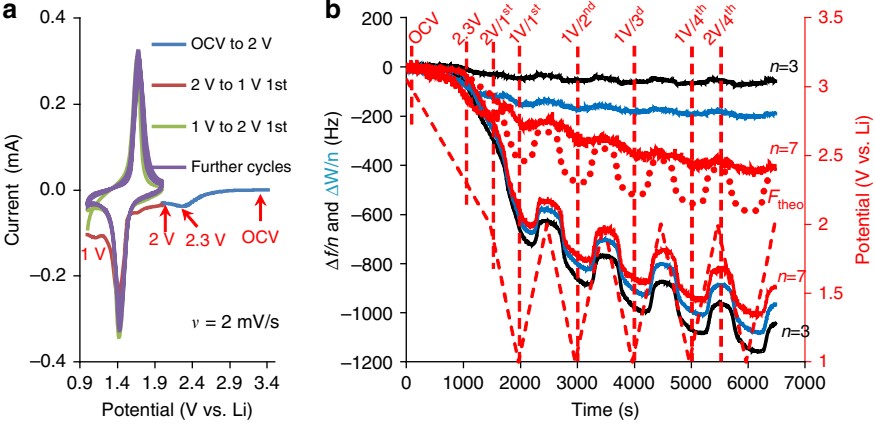

**Fig. 3** Characterization of LTO electrodes in EC-DMC 1 M LiPF$_6$ solutions. **a** Two first sequential CVs from OCV to 1 V and then to 2 V, as indicated. **b** Frequency and resonance width changes at different overtones (shown by different colors) as functions of time during electrode cycling. The dashed red lines denote the potentials at which viscoelastic analysis has been carried out

and $\Delta f/n$ were fitted to the viscoelastic model. The quality of the fit can be seen in Supplementary Fig. 2a, b, whereas the error bars characterizing precision of $\Delta f/n$ and $\Delta D$ shifts in the solution with respect to the frequency and dissipation of the neat crystal in the air are shown in Supplementary Fig. 2c, d. We have found that the first intercalation of Li-ions into a virgin LTO electrode occurs directly, without mediation by surface films, because the potential for SEI formation has not yet been reached. In this potential region, $\Delta f/n$ is $n$-independent and practically equal to $F_{theo}$ reflecting the Sauerbrey's mass of the inserted Li-ions. $F_{theo}$ was calculated from the intercalation charge combining the Faraday law and Sauerbrey equation. Such calculations are valid since the change in $\Delta D$ is negligible.

In addition, we note that surface films are formed almost completely throughout a short period of potential scan (lapsed for 350 s), starting from 1.35 V during the first scan, reaching a vertex potential of 1 V, and then continuing in the anodic scan up to a potential of 1.35 V (Fig. 2b). Only in this potential range, the amplitude of $\Delta f/n$ and $\Delta D$ changes strongly depends on $n$, corresponding to a very small irreversible capacity, expressed by the difference in the CV current during the first and the subsequent potential scans (see the solid and the dotted lines, respectively). During the subsequent cycling, the relative changes of $\Delta D$ and $\Delta f/n$ become almost $n$-independent (Fig. 2c). Taking into account that the Sauerbrey-mass effect due to the inserted/extracted Li-ions essentially matches the experimental $\Delta f/n$ shift, and the potential-dependent changes of $\Delta D$ are small, only a minor change in the viscoelastic properties of the electrode layer takes place (Supplementary Table 1). Most importantly, the parameters of the SEI in the neighboring intercalation and deintercalation states remain the same (see the data for states 1 and 2 V for the seventh cycle in Supplementary Table 1), probably because of the absence of the intercalation-induced volume change of the LTO electrodes.

**Surface films on LTO electrodes in LiPF$_6$-containing solutions.** The formation of surface films on LTO electrodes in the LiPF$_6$ solution seems to be drastically different from that in LiTFSI solutions, as can be seen from the related CVs (Fig. 3a) and from the $\Delta D$ and $\Delta f/n$ changes (Fig. 3b). A large irreversible capacity can be clearly seen during the first scan, starting from 2.6 V, passing through a small peak at 2.3 V and then accompanying the first Li-ion insertion at 1 V. The irreversible capacity is still visible during the anodic scan toward 2 V. In this potential range of the first scan, with an offset for $\Delta D$ and $\Delta f/n$ at OCV, the $n$-

dependence of both these quantities is the strongest. After completing the first cycle at 2 V, further cycling provides potential-dependent changes of $\Delta D$ and $\Delta f/n$ (as expected), which become essentially $n$-independent. This implies that Li-ion insertion/extraction into/from the electrodes occurs across the stable SEI-type surface films. Like with LiTFSI solutions, fitting the $\Delta D$ and $\Delta f/n$ changes to the viscoelastic model was performed at eight different potentials, designated in Fig. 3b by vertical dashed red lines. The change in $\Delta f/n$ for LiPF$_6$ + 2% VC solution is qualitatively similar to that for the additives-free LiPF$_6$ solution, although the background line for $\Delta f/n$ decreases more steeply than that for the LiPF$_6$ solution. This implies the continuous growth of the surface films for a longer period of time compared to that in the neat LiPF$_6$ solution, due to the presence of the surface reactive VC additive (Supplementary Fig. 3). In addition, at relatively high potentials, around 2 V, we have observed large peaks in the change of dissipation, revealing that the surface films are formed under different conditions than those related to the neat LiPF$_6$ solution. The entire sets of parameters for the neat LiPF$_6$ and LiPF$_6$ + 2% VC solutions are listed in Supplementary Tables 2 and 3, respectively. The quality of the fit for each of these two electrolyte solutions can be seen in Supplementary Figs. 4 and 5, respectively.

**Quantification of the viscoelastic parameters.** As can be seen from the sketch of the velocity profiles in Fig. 1f, the model provides the characteristic parameters of two acoustic layers of the composite electrodes, the SEI and the semi-infinite Newtonian liquid in contact with the electrode surface. Whereas the specific density and viscosity, $d_{sol}$ and $\eta_{sol}$, of both the LiTFSI and neat LiPF$_6$ solutions, remain constant throughout the entire cycling tests, the presence of VC in LiPF$_6$ results in a clear 5% increase in $\eta_{sol}$ at about 2.2 V, where the change in $\Delta D$ after a few cycles is the maximal one and is above the zero line of the ordinate. This increase in viscosity indicates the expected polymerization of VC occurring during the first two cycles.

As has been earlier mentioned, our strategy was focused on finding the best conditions for the experiments which link the experimental changes of $\Delta f/n$ and $\Delta D$ to the formation and growth of SEI rather than to viscoelastic changes in the electrode bulk. In this respect, most important is the selection of an optimal loading mass density of the electrode coating: a too heavy electrode (too thick coating) is seen by EQCM-D as exceedingly soft. The measurements are accompanied by reduction of the number of the measured overtone orders. In contrast, selection of

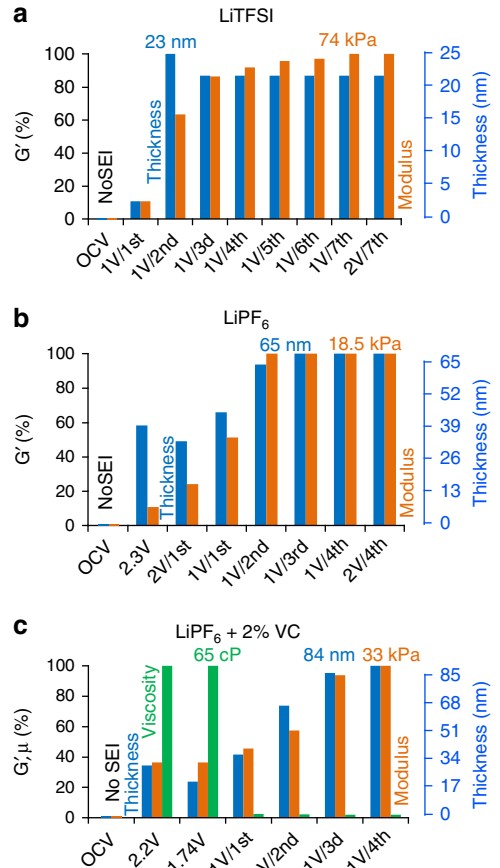

**Fig. 4** Viscoelastic parameters of LTO electrodes when in contact with different electrolyte solutions: **a** 1 M LiTFSI, **b** 1 M LiPF$_6$, and **c** 1 M LiPF$_6$ + 2% VC. Storage modulus and thickness of the SEI on LTO electrode at different potentials are shown by orange and blue columns, respectively. The VC green columns show the dynamic viscosity of SEI. The full set of parameters for all three systems is listed in Supplementary Tables 1, 2 and 3, respectively

an extremely small loading mass (i.e., too thin and most probably non-continuous coating) is unfavorable because of the large "floodness factor," $S_{flood}$, defined as the ratio of the mass of the solution to the mass of the electrode, enhancing the continuous parasitic reactions (see the next section).

As a typical example, we consider LTO coating with a mass loading of 88 μg/cm$^2$ (the frequency shift in air 5 kHz) with an average thickness of 420 nm (measured by AFM). If this electrode coating would be completely rigid in air, the change of the dissipation factor for all overtone orders should be zero with respect to that of the neat quartz crystal. In this case, EQCM-D would "see" the coating as a single rigid layer despite the fact that physically it consists of a few or more individual layers of intercalation particles. However, experimentally, the values of $\Delta D$ are finite rather than zero and, in addition, they depend on $n$. The application of the viscoelastic model to the dissipation changes allows to reliably retrieve the viscoelastic parameters of the electrode's active mass layer because the change in $\Delta D$ does not depend on the inertial loads (processes), whereas $\Delta f/n$ changes occur due to both viscoelastic and inertial loads. This is the most important essence of our approach to the viscoelastic analysis of the particulate composite electrodes in contrast to conventional continuous uniform viscoelastic films.

If the viscoelastic layer would be the single one, then the $\Delta f/n$ calculated with the parameters obtained from fitting the $\Delta D$ should ideally match the experimental frequency changes.

However, if an approximately $n$-independent shift is observed between the fitted and the experimental values of $\Delta f/n$, this should be ascribed to an effect related to the inertial loading. Usually the nature of the additional inertial loading affecting $\Delta f/n$ rather than $\Delta D$ is identified from the information obtained by the supplementary techniques, very often from SEM images of the electrode coatings. In the considered experiment in the air, this implies the existence of completely rigid (rather than viscoelastic) electrode layer. Naturally, the mass contribution of this rigid bottom layer is due to the difference of 3.25 kHz frequency shift as shown in Supplementary Fig. 1b. Confirmation of the rigid character of the bottom layer comes from the SEM images revealing a non-uniform distribution of the binder: whereas the viscoelastic layer is located at the top of the electrode (facing the electrolyte solution) less strongly bound to the neighboring intercalation particles, the bottom layer is more strongly embedded into the binder network, attaching it to the current collector.

When this coated electrode is immersed into the solution (Supplementary Fig. 2), the application of viscoelastic model shows that the parameters of the both layers are not appreciably changed, and we find the frequency change between the fitted and the experimental values equal to 5.2 kHz, i.e., close to 5 kHz (the difference 200 Hz can be related to the impregnation of the narrow pores of the rigid layer). Since the viscoelastic parameters do not appreciably change with potential and time, the potential and time dependence of $\Delta D$ is ascribed to the formation of SEI, whereas the related changes of $\Delta f/n$ may reflect in addition some inertial processes such as insertion of electrolyte solution into narrow pores of rigid bottom layer, or the change of the porous structure with time. This explains why the time-dependent dispersion of the experimental values of $\Delta f/n$ is larger than that calculated from the model: note that this dispersion does not affect the assessment of the viscoelastic characteristics of the SEI from the dissipation changes but only indicates the presence of additional processes (inertial loads) related to the interaction of the bottom layer of the electrode with the electrolyte solution (Supplementary Fig. 2).

The SEI thickness and shear-storage modulus obtained for all three Li-battery electrolyte solutions are presented in Fig. 4, which portrays the dynamic formation and growth of the SEI films. For the LiTFSI solution, the formation of the SEI starts after the first Li-ion intercalation and is almost completed in the second cathodic scan, after reaching the potential of 1 V. The SEI thickness stabilizes at about 20 nm (Fig. 4a), in agreement with the electrochemical data in Fig. 2a, and SEM image in Fig. 1b. The consecutive cycling of LTO electrodes in this solution results in the rapid growth and further stabilization of $G'$ at a level of 74 kPa. In contrast, the formation of the SEI in the neat LiPF$_6$ solution starts at much higher potentials (before the first Li-ion intercalation) and SEI growth continues during the first two cycles. The thickness of this SEI is 3.2 times larger and the storage modulus is 4 times smaller than that of the SEI formed in the LiTFSI solution. The dynamics of formation of the SEI in the presence of VC is really peculiar: unlike the case of neat LiPF$_6$ and LiTFSI solutions, the initial formation of the SEI in the presence of VC is characterized by a very high solid-state viscosity (65 cP), implying the participation of oligomeric species of VC in the growing SEI. During the subsequent cycling, the SEI is reorganized, its viscosity drastically decreases and the film becomes thicker and stiffer during subsequent cycling. As a result, $G'$ of the SEI formed in the presence of VC has intermediate values between those of the SEI formed in LiTFSI and LiPF$_6$ solutions (33 kPa), and this SEI is the thickest (84 nm) among all three SEI films formed on LTO electrodes in the three solutions we explored.

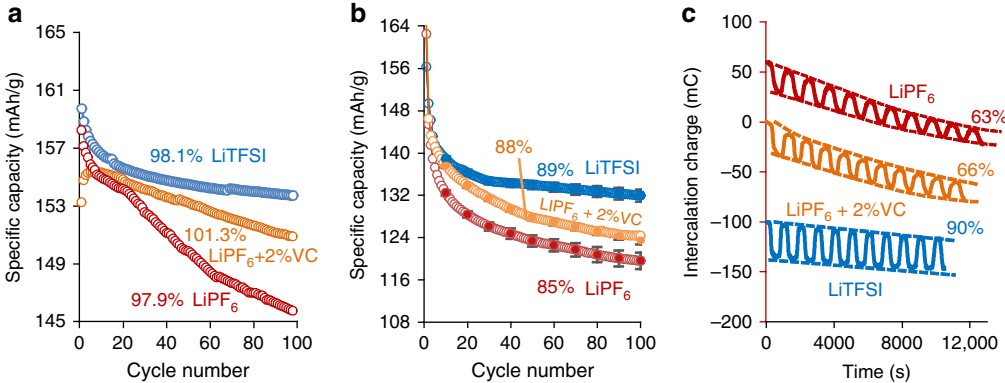

**Fig. 5** Cycling experiments of LTO electrodes in cells with different floodness factor, $S_{flood}$. Cycling in coin cells at 1C rate **a**, **b** and in experimental EQCM-D cell (scan rate 2 mV/s) in three electrolyte solutions as indicated. Capacity retention for LTO electrodes in all three solutions are referred to 10th cycle and expressed as % with respect to the capacity during the first charge. The ratio of LTO mass to the mass of the solutions were equal to: **a** 5 mg LTO/50 mg solution ($S_{flood} = 10$); **b** 80 µg LTO/50 mg solution ($S_{flood} = 6.25 \times 10^2$); **c** 90 µg LTO/1 g solution ($S_{flood} = 1.10 \times 10^4$). For clarity, in **c** the intercalation charge is shifted by 60 and −100 mC for LiPF₆ and LiTFSI solutions, respectively

There is no doubt that the primary reason for the different intrinsic mechanical properties of the SEI formed in the different electrolyte solutions is the chemical nature of the related surface species, caused by the reactions of the LTO electrodes' surface with the solution components. In this respect, it is important to examine the surface composition by XPS and find evidence that the nature of the surface species formed in our experimental EQCM-D cells under typical flooded conditions is similar to that reported in the literature for LTO electrodes cycled in coin cells[6,7,10] (Supplementary Fig. 6). For LiTFSI solutions, the related SEI is enriched in $(-CH_2-CH_2-O-)_n$ moieties due to reduction of EC, which leads to ring opening. The SEI contains ROLi, $R-CH_2-OCO_2Li$ species and $Li_2CO_3$, (reduction products of the alkyl carbonate solvents) along with the decomposition product of the LiTFSI salt. The content of LiF (due to reduction of the TFSI anions in the presence of Li-ions) is much smaller than that found in the SEI formed on LiPF₆ solution. A mixture of different C–O and C = O species, as well as species with a fluorinated carbon bond, are found as well. Most importantly, two Ti 2p core peaks are located at the same binding energy for the films formed in both salts, but the Ti atomic concentration is higher (by a factor of 6) for SEI formed in LiTFSI solutions than for SEI formed in LiPF₆ solutions, because in the latter case the surface films mask the Ti response, in qualitative agreement with the results of the viscoelastic modeling and SEM images shown in Fig. 1. According to the literature data, oligomeric/polymeric species of VC are the additional component of the SEI formed in VC-containing solutions, compared to the SEI formed in the neat LiPF₆ solution.

However, the chemical nature of the SEI formed in the different solutions is not the only reason determining their mechanical properties. It is reasonable to assume that the floodness factor, $S_{flood}$, affects the SEI mechanical properties. This factor directly determines the extent of irreversible surface reactions, which, in turn, influences the SEI thickness and porosity and, eventually, may control the electrode capacity retention during cycling.

The EQCM-D methodology is internally limited to a small-load approximation (SLA)[25], requiring some excess of electrolyte solution, and there is no other in situ technique that can probe the viscoelastic layers in a non-invasive shear mode in practical Li-ion cells. Hence, the major question is whether the intrinsic mechanical properties of the SEI-type surface films on LTO electrodes assessed by EQCM-D for different electrolyte solutions, are also characteristic of the surface films formed on practical composite electrodes, like those usually cycled in pouch or coin cells.

**Effect of floodness factor on the SEI quality**. We have approached this important problem indirectly, by comparing the capacity retention of LTO electrodes during cycling experiments in coin cells to that observed in cycling experiments with our research EQCM-D cells. The floodness factor, which we mark as $S_{flood}$, is not the same for all the cells (see Fig. 5). Obviously, when the $S_{flood}$ increases, first by a factor of 63 (from Fig. 5a–b) and then by 3 orders of magnitude (for the EQCM-D cell), the absolute values of the capacity retention decrease accordingly (the capacity retention at the end of the 10th cycle in Fig. 5a, b for coin cells, and at the end of 20th cycle for EQCM-D experiment in Fig. 5c is indicated). The practical specific capacities of the three electrode coatings onto the quartz-crystal surfaces in the three different electrolyte solutions are indicated in Supplementary Fig. 7a together with the related CVs measured during the second cycle. Moreover, we have prepared LTO coatings on glassy carbon disk electrodes with a small diameter (1 mm) with the estimated active loading mass of 1.7 µg: $S_{flood}$ of this system was 52 times larger than that observed for the electrode coatings in the EQCM-D measurements, see Supplementary Fig. 7b. It is clearly seen that independent of the value $S_{flood}$, the sequence of the values of capacity retention in the three different electrolyte solutions in the research EQCM-D cells is exactly the same as that obtained for coin cells with two different values of $S_{flood}$ after 100 cycles. With the glassy carbon current collector of the smallest surface area, the electrode showed the highest specific capacity in the LiTFSI solution. However, the largest floodness factor of $5.7 \times 10^5$ seriously deteriorates the reversible electrode capacity in the sequence LiTFSI«LiPF₆ + 2% VC < LiPF₆. Note that the extent of the parasitic cathodic reactions, both on glassy carbon electrode and on glassy carbon with electrodeposited Cu layer, were relatively small (inset in Supplementary Fig. 7), thus the deterioration of the reversible capacity of LTO electrodes in LiPF₆ solutions is due to the catalytic character of the parasitic reactions on the LTO surface.

The correspondence between $S_{flood}$ and the extent of irreversible capacity affecting the absolute values of capacity retention is not surprising. To some extent, it resembles the effect of elevated temperature on the irreversible capacity of intercalation electrodes. The closest analog of this phenomenon is the time–temperature correspondence principle in viscoelastic polymers where their complex frequency-dependent shear modulus measured at low temperature appears to be equal to the modulus acquired at a higher temperature and a higher frequency (or shorter time). The physical reason for the $S_{flood}$–temperature correspondence is, of course, the lower quality of the protecting

SEI in the electrodes, resulting in higher irreversible capacities when either the temperature or $S_{flood}$ is increased. The physical properties of the SEI that may suffer pronouncedly as $S_{flood}$ increases are, in our opinion, its porosity and thickness. The SEI pores (filled with electrolyte solution), which are smaller than the length of the shear wave, are effectively included in the SEI viscoelastic characteristics assessed by EQCM-D, resulting in lower effective elastic moduli. Thinner SEI films (formed in LiTFSI solution) having larger elastic modulus are therefore considered in our EQCM-D approach as having higher quality compared with thicker porous SEI films (formed in $LiPF_6$ solution). This results in a decrease of the elastic shear modulus as seen from the diagrams in Fig. 4. The case of the VC-containing solutions is very interesting, as the initial increase in capacity retention for this solution in coin cells is linked to a larger irreversible capacity in a few initial cycles. However, during subsequent cycling, the quality of SEI improves and the extent of irreversible capacity decreases, with a simultaneous increase in the capacity retention. The transient behavior of the SEI formed on the surface of LTO in the presence of VC in the related EQCM-D test, i.e., a decrease in the initial SEI viscosity and much slower increase of thickness and shear-storage modulus with cycle number (Supplementary Fig. 3 and Fig. 4c) compared to that for the two other solutions is in a qualitative agreement with the transient behavior in the related coin-cell (Fig. 5a) and also with the literature data related to high-precision coulometry tests[19].

## Discussion

We have developed a highly sensitive EQCM-D-based methodology for in situ assessment of gravimetric and viscoelastic changes in composite LTO electrodes caused by simultaneous Li-ion insertion/extraction and the formation/growth of SEI on their surfaces in three different Li-battery electrolyte solutions. The acoustic multilayer formalism was used to build a self-consistent viscoelastic model of composite LTO electrodes, describing their experimental resonance frequency and resonance width changes in gas/air environment, in contact with Li-battery electrolyte solutions at OCV, and under applied potential. The model selectively characterizes the mechanical state of each layer in the multilayer composite electrodes assembly (rigid and viscoelastic parts of the electrode + SEI) in contact with the electrolyte solution. EQCM-D probes electrodes viscoelasticity in non-invasive, thickness-shear resonant mode, and in real time. The gravimetric and viscoelastic parameters of the layers are obtained via fitting an advanced viscoelastic model to the experimental EQCM-D data. Our major finding is that Li-battery electrolyte solutions and additives can be preliminarily screened in EQCM-D short-time experiments. Simultaneously with probing the electrode capacity retention, we have quantified intrinsic viscoelastic characteristics of the growing SEI films. Our conclusion is that using the moderately flooded research EQCM-D cells may provide important predictive information about the cycling performance of practical electrodes tested in coin cells in various electrolyte solutions. The developed experimental setup and modeling routine can be applied to a large variety of all kinds of ions insertion electrodes, protected by SEI-type surface films, including low-voltage anodes and high-voltage cathodes. We are currently focused on reducing $S_{flood}$ in improved research EQCM-D cells by one order of magnitude, which is enough to open the door to a very broad application of EQCM-D in advanced energy-storage research.

## Methods

**Materials and in situ experiments**. $Li_4Ti_5O_{12}$ powder (from MTI Corp (USA)) was ball-milled into fine powder (average particle size ca. 100 nm). A slurry contained 80% LTO, 10% PVdF binder (Kynar HSV 900), 5% Super-P carbon black, and 5% CNT (BAYTUBES C150P, Bayer MaterialScience) in N-methyl

pyrrolidone of a suitable viscosity was prepared. This slurry was spray-coated on a Cu-covered quartz-crystal surface (purchased from Q-Sense-Biolin Scientific AB, Sweden) placed on a hot plate (the temperature was kept at 150 °C), thus forming composite electrodes with thin Cu current collectors. After drying at this temperature for 1 h, the electrodes were slowly cooled down to room temperature, put into EQCM-D cells and measured, first in air and then in the desired solutions at OCV and under applied potential (during the electrochemical processes).

For coin-cell measurements, the same components were mixed with N-methyl pyrrolidone and coated on Cu foils by the doctor-blade method. The foils with the slurry were dried for at least 4 h under vacuum $10 \times 10^{-3}$ torr at 100 °C, then punched and weighted. Dried LTO electrodes were kept in a glove box filled by highly pure argon. We prepared two batches of electrodes, a batch of thick electrodes with LTO loading of 4.5–4.8 mg/cm², and a batch of thin electrodes with a loading of 0.04–0.08 mg/cm². Thin electrodes were coated on hot Cu foils by the airbrush method from a diluted dispersion. Before coating, the dispersion was sonicated to prevent particle agglomeration.

Three electrolyte solutions were used: 1 M $LiPF_6$/EC:DMC (1:1 v/v) from Merck KGaA, Germany, 1 M $LiPF_6$/EC:DMC (1:1 v/v) with 2% wt vinylene carbonate from Solvonic, France and 1 M LiTFSI/EC:DMC (1:1 v/v) from Solvonic, France. The amount of water in all electrolytes was ≤ 20 ppm.

**Measurements in coin cells**. For the electrochemical measurements, we used two-electrode coin-type cells (2325, NRC, Canada), a polyethylene (Celgard) separator and 50 μl of electrolyte solution. The cells were assembled in an Ar-filled glove box ($O_2$ level < 10 ppm, $H_2O$ level < 1 ppm). All cells were cycled using Arbin BT2000 battery-testing multichannel systems (Arbin Instruments, USA). LTO/Li half-cells were cycled with cutoff limits of 2–1 V vs. $Li/Li^+$. The high mass and sprayed coin cells were cycled at rates of C/2 and 2C, respectively.

**Measurements in EQCM-D cells**. Multi-harmonic quartz-crystal measurements using EQCM-D were performed with a Q-Sense E1module (QCM-D from Bioline Scientific) at overtone orders of $n$ from 3rd to 13th (higher harmonics were not measured for thick extensively viscoelastic electrodes). The volume of the solution above the AT-cut 5-MHz crystal was about 1 ml. The temperature was kept constant at 22 °C. The data acquisition was performed using the QSoft401 software. In addition to QCM-D, electrochemical measurements were performed by means of a Bio-Logic VSP-300, using the EC-Lab software. The electrochemical potentials were measured and reported vs. the Li reference electrode. The counter electrode was thin Li foil pressed onto Ni mesh.

**HR-SEM imaging**. The images of pristine and cycled electrodes were obtained using Magellan 400 L (FEI) scanning electron microscope. Cycled electrodes were transferred from the glove box into the microscope chamber using a homemade hermetic transfer cell in order to prevent further oxidation of the electrode surface.

**Additional experimental and modeling details**. Viscoelastic Voigt-type models (Supplementary Note 1), Optimal electrode mass density for sensing SEI formation (Supplementary Note 2), Statistical errors of EQCM-D measurements (Supplementary Note 3).

**Data availability**. The data that support the findings of this study are available from the corresponding authors on reasonable request.

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

## Acknowledgements

N.S. thanks the Israel Ministry of Science Technology and Space for financial support.

## Author contributions

V.D., N.S., and S.S. carried out in situ EQCM-D measurements and obtained HE SEM images, P.N. performed synthetic work and optimization of fabrication of slurry for airbrush coating, L.D. carried out viscoelastic modeling, whereas D.A. and M.L. conceptualized the entire work.

## Additional information

**Competing interests:** The authors declare no competing financial interests.

