## [Peer Review File · Nature Communications]

Reviewers' comments:

Reviewer #1 (Remarks to the Author):

General comment:

This manuscript describes some rheological properties of the SEI layer formed on LTO electrodes. It is an original work and at the best of my knowledge it is the first time that this can of study it is run by using an in-situ technique. It is clear for me that this work will be interesting and useful mainly for the lithium-ion battery community; however, considering the QCM-D use, it could be also interesting for a larger community.

Some technical questions:

1- I think that authors could clarify why they use the Voigt model in some specific potentials.

2- In figure 2, they use the charge to calculate the change in frequency associated to lithium intercalation; but, in figure 3 they have not shown the same data (change in frequency due to lithium intercalation). I think that this comparison can be useful to improve the conclusions related to the irreversible capacity.

3- It is interesting that lithium intercalation does not produces important changes in the energy dissipation. Does it mean that there is not produces important volumetric changes in the LTO materials?

Conclusions:

I think that is a very nice work; however, I am not convinced that it could be interesting for a wider community.

Reviewer #2 (Remarks to the Author):

This paper is interesting and presents newly quantifiable properties of common SEI features. The choice of materials and electrolytes is pertinent to research and industry needs. It is especially useful that you link your results to the chemical nature of the electrolyte differences. Figure 2 is particularly well done. As you'll see in the detailed comments below, certain clarifications are needed for this work to be more understandable and reproducible.

1 - The paper would benefit from an English grammar edit. For example, the first sentence of the introduction dissuades one from reading further based on poor subject-verb agreement and complicated structure.

2 - In the first sentence of the abstract, SEI is not spelled out properly. As you know, S stands for solid, not surface.

3 - I strongly believe that the carbon-based binder system should be analyzed without LTO on the surface of the sensor. What is its contribution to the viscoelasticity measured? This is especially pertinent because the thicknesses of the cathode layers on the sensor are NOT the same for each sample (quite different in one case).

4 - Why was the thickness of the LTO for one electrolyte so much thinner than the others? This should be addressed. This also calls attention to the fact that this work seems like it was never reproduced in-house.

5 - This is data from CV performed on three different surfaces, one time each. Although the results make sense to anyone who has worked with these materials, how reliable can the quantification be

if this was only done once per electrolyte? I find the lack of error bars on this data very troubling. What would happen if the samples were thinner or thicker, or the experiments simply performed on a different day?

6 - You describe a rigid and a less rigid layer for the LTO and explain that this difference is based on the distance from the sensor. What are the two thicknesses? In the schematic in Figure one they appear to be equal thicknesses. This should be clarified.

7 - I see no addressing of the impact of electrolyte inclusion into the pores of the structure. This is important because the porosity of the electrodes is not reported, and the electrodes are of varying thicknesses. Address how this factor affects your results and conclusions. Keep in mind that the electrolytes have different wetting abilities.

8 - What is the LTO particle size after ball milling?

9 - CV does not imply that all of the LTO reacts with lithium (unlike a slow galvanostatic experiment). What is the actual amount of lithiation and delithiation based on the coulometry? This is needed information because the reader wants to know if the "rigid" LTO is even being lithiated/delithiated during the CV.

10 - References are needed for the sentence in lines 120-121. I have seen surface oxides lithiate at 1.5 V in the literature, and the three electrolytes have different onset SEI formation potentials. References needed.

11 - In line 192, it seems odd to call SEI formation in the presence of VC "impressive". That's a qualifying statement that detracts from the scientific nature of this work. Do you mean to imply that this SEI is the best?

12 - Address if the 150 degree treatment of the electrodes has any negative effects on the binder. It's notable that for the coin cell electrodes you only went to 100 degrees. That's because 150 degrees can be too hot for this binder system.

13 - How was the water level in the electrolyte determined?

14 - In the EQCM, was Li the counter electrode and reference? If so, how was it adapted into the cell. Was the counter/reference something else, a pseudo? If so, what was it?

15 - Line 367, you mean non-aqueous.

16 - Figure 4 needs a thickness axis. Writing the number in nanometers is not enough. The bars being shorter for the VC case lead to an erroneous first interpretation. Axis needed.

Response letter to the reviewers' comments on our paper NCOMMS-17-07897A-Z, entitled: "In Situ Real-Time Gravimetric and Viscoelastic Probing of Surface Films formation on Li Batteries Electrodes by EQCM-D: Composite Li₄Ti₅O₁₂ Electrodes as a Typical Example".

We received two, very helpful, reviewers' reports.

We copied below all the reviewers' comments and below each of their comment we provide our response in red explanations how the paper was revised in light of each comment. The corrections made in the revised MS are marked in yellow.

Reviewer #1 (Remarks to the Author):

General comment:

This manuscript describes some rheological properties of the SEI layer formed on LTO electrodes. It is an original work and at the best of my knowledge it is the first time that this can of study it is run by using an in-situ technique. It is clear for me that this work will be interesting and useful mainly for the lithium-ion battery community; however, considering the QCM-D use, it could be also interesting for a larger community.

Authors' response: We wish to cordially thank this reviewer for the time devoted to reading of our manuscript and very valuable advices.

Some technical questions:

1- I think that authors could clarify why they use the Voigt model in some specific potentials.

Authors' response: The goal was to establish whether the intercalation level of LTO electrodes affects the mechanical properties of the SEI. This would be expected if the intercalation-induced changes of the electrodes' volume are large. In view of the lack of potential-induced volume changes in LTO electrodes, virtually unchanged parameters of the SEI for completely intercalated and completely deintercalated states are consistent with the zero-volume change condition. We added one sentence in this respect in p. 7.

2- In figure 2, they use the charge to calculate the change in frequency associated to lithium intercalation; but, in figure 3 they have not shown the same data (change in frequency due to lithium intercalation). I think that this comparison can be useful to improve the conclusions related to the irreversible capacity.

Authors' response: The reviewer is right. We have corrected the figure.

3- It is interesting that lithium intercalation does not produce important changes in the energy dissipation. Does it mean that there is not produce important volumetric changes in the LTO materials?

Authors' response: Yes, this is exactly the situation. This material is considered as zero-stress material, with negligibly small intercalation-induced volume changes. In fact, it is for this reason that we selected this electrode material for our viscoelastic analysis of the SEI forming on its surface. In this work and related paper we describe a new methodology based on QCM-D techniques + relevant modelling, to follow an understand SEI type surface films formation on Li ion insertion electrodes. For a first demonstration, we have chosen a zero-stress electrode material, so we can concentrate more on the intrinsic surface chemical/electrochemical effects. The success of this study enables to go further and explore more complicated situations in which surface films are formed on intercalation electrodes which volume changes as a function of the intercalation process.

Conclusions:

I think that is a very nice work; however, I am not convinced that it could be interesting for a wider community.

Authors' response: The community involved in the energy storage problems is probably the largest in material science field. However, since we developed herein general analytical tools + relevant modeling, this paper can be interesting for researchers from the other fields: catalysis, polymer science, corrosion (i.e. where high precision acoustic sensors are used). Thereby, we are confident that this paper and the related work could potentially interest a broad readership of this journal.

Reviewer #2 (Remarks to the Author):

This paper is interesting and presents newly quantifiable properties of common SEI features. The choice of materials and electrolytes is pertinent to research and industry needs. It is especially useful that you link your results to the chemical nature of the electrolyte differences.

Figure 2 is particularly well done. As you'll see in the detailed comments below, certain clarifications are needed for this work to be more understandable and reproducible.

Authors' response: We are very grateful to this reviewer for his very careful reading of our paper and his very helpful comments.

1 - The paper would benefit from an English grammar edit. For example, the first sentence of the introduction dissuades one from reading further based on poor subject-verb agreement and complicated structure.

Authors' response: We deeply apologize for this unfortunate circumstance. It is now corrected and the grammar check has been done over the entire text.

2 - In the first sentence of the abstract, SEI is not spelled out properly. As you know, S stands for solid, not surface.

Authors' response: Thank you, this technical fault has been corrected.

3 - I strongly believe that the carbon-based binder system should be analyzed without LTO on the surface of the sensor. What is its contribution to the viscoelasticity measured? This is especially pertinent because the thicknesses of the cathode layers on the sensor are NOT the same for each sample (quite different in one case).

Authors' response: The reviewer is absolutely right. The origin of viscoelasticity of the composite electrodes almost always relates to the mechanical properties of the binder. Sharing the reviewer's opinion on the possible role of the binder in determining composite electrodes' viscoelasticity, we have performed special experiments with neat PVdF films attached to quartz crystal surfaces. The coated crystals were immersed in an aqueous 1M Li_2SO_4 solution in one case, and in 1M LiPF_6 in EC+DMC, in another case. The normalized frequency and resonance width changes are presented as functions of the penetration depth (the frequency shift was corrected for the contribution due to the dry mass of the film). Such a plot is convenient for distinguishing ideally stiff materials from the viscoelastic ones. If the immersed film is dense and has a flat surface, then the frequency and resonance width changes are exactly the same as those measured from the uncoated neat crystal. This is because a dense coating with flat surface dissipates oscillation energy and contributes to the frequency change exactly like a rigid flat surface in contact with the same liquid. PVdF in 1M Li_2SO_4 solution behaves exactly in such a way.

PVdF in 1M LiPF_6 in EC+DMC reveals a small increase in resonance width and a much larger decrease in the frequency change. This indicates that the binder is slightly

viscoelastic, and in addition, absorbs some electrolyte solution. But it is certainly not extensively viscoelastic, and its viscoelastic properties do not depend on the time of storage typical for measurements of LTO composite electrodes. Hence we believe that its slight viscoelasticity does not compete with the pronounced viscoelastic changes due to the formation and growth of SEI type surface films (see the figure below). We prefer to complete the entire characterization of the neat binder coating and then compare it with storage and cycling experiments of the composite electrodes having large intercalation-induced volume changes. Our answer to the comment on the choice of the electrodes thicknesses for the different electrolytes is given elsewhere in the present authors' response.

Normalized $\Delta f/n$ and $\Delta W/n$ changes after immersion of a thick neat PVdF film (mass loading density $80 \mu\text{g}/\text{cm}^2$) in aqueous solution of 1 M Li₂SO₄ and in aprotic solution of 1 M LiPF₆ in EC+DMC (rhombs).

4 - Why was the thickness of the LTO for one electrolyte so much thinner than the others? This should be addressed. This also calls attention to the fact that this work seems like it was never reproduced in-house.

Authors' response: We thank the reviewer for this most important remark. Unfortunately, we have done a triple fault preparing this manuscript: (i) the values of thickness or mass density of the electrodes were not indicated, (ii) the origin of the shift between the experimental and the calculated values of $\Delta f/n$ was explained for measurements in the air only (Fig. S1); similar shifts in the solutions (Figs. S2, S4 and S5)

have not been discussed in detail; from the differences in the shifts in Figs. S2, S4, S5 the reviewer has concluded that the electrodes possessed significantly different thicknesses, (iii) the choice of the electrode thickness for measurements in different electrolyte solutions were not clarified. Finally, the issue of precision and repeatability of the measurements was not addressed. Most part of the revision of the main text and Methods section relates to the above 4 issues + a better description of a non-conventional use of the viscoelastic model applied to the particulate composite electrode coatings.

The frequency shift in the air due to the “dry” electrode mass density for the measurements in LiTFSI, LiPF₆ and LiPF₆+2% VC were 5.0, 4.7 and 3.6 kHz corresponding to the following loading mass densities: 88, 83 and 64 μg/cm². A standard precision of QCM-D instrument is ±0.1 Hz. However, the precision of fabrication of dry electrode coatings is limited by the use of the manual airbrush spraying technique. From 3-6 attempts the best precision we could typically reach was ± 15 Hz, for example, the frequency change for dry LTO coating further tested in LiTFSI solution was 5000Hz ± 15 Hz.

The dry mass of the coating is obtained from the difference of the frequency of the neat (uncoated) crystal from which the frequency of the coated crystal is subtracted. Here another limitation of the precision related to the different static stresses on the crystal during its mounting to the cell via o-ring is approached. The measurement of the frequency change of the dry coating implies disassembling of the cell after measurement of the neat crystal, and then its further assembling for measurement of the coated crystal. Reassembling of the cell typically results in a random frequency shift of the same order of magnitude as the electrode fabrication, ±15 Hz. If reassembling of the cell is not involved, e.g. deposition of thin metallic films in a vacuum takes place, the precision increases by 2 orders of magnitude.

Let us consider the precision of frequency measurements of a dry coating after its immersion into liquid (electrolyte solution). For example, for the 5 kHz (frequency shift in air) LTO electrode coating obtained on 3rd overtone, after its immersion into a LiTFSI solution, the frequency shift with respect to that of the neat crystal in the air for 3-4 samples is equal to 8240 Hz ± 20 Hz. This good precision was reached only after the introduction of impregnation of the coated crystal in solution under vacuum (as is usually done for thick electrodes for measurements in coin cells). Assuming that reassembling of the cell ensures the precision of ± 15 Hz, the minor remaining contribution, ± 5 Hz, should be assigned to the variation of the porous structure of the different samples, possible swelling, and other minor effects. Error bars are indicated in Fig. S2c, d.

We now clarify the issue of unequal loading masses for 3 types of solutions. This is related to the fact that due to the different viscoelastic properties of the composite electrodes in different solutions, the number of measured overtones starts to depend

on the electrode thickness. On the one hand, the increasing (electrolyte-dependent) viscoelasticity of the electrode raises the dissipation to a large extent, especially on the higher harmonics. This implies an inability to clearly observe the resonance in the electrode using high harmonics, only lower harmonics can be measured. For viscoelastic analysis, the measurements for at least three different harmonics are required. With LiTFSI we were able to measure the frequency and dissipation changes in 5 kHz LTO electrode ($88 \mu\text{g}/\text{cm}^2$ mass density) with the use of five harmonics. However, even decreasing the mass density to $83 \mu\text{g}/\text{cm}^2$ and $63 \mu\text{g}/\text{cm}^2$ for LiPF_6 and $\text{LiPF}_6 + 2\% \text{ VC}$, respectively, we were able to measure only three overtone orders (minimal number required for viscoelastic analysis). This information is included in the text of the revised paper in detail. Moreover, we show that even for the best electrolyte solution, i.e. LiTFSI, a larger loading mass of $105 \mu\text{g}/\text{cm}^2$ (instead of $88 \mu\text{g}/\text{cm}^2$) reduces the number of the measured harmonics to two (see Fig. S7 and its discussion in the text).

From the above result, one can suggest that it is preferable to decrease mass densities of the electrodes in order to measure the maximal number of harmonics. Yes, in this case, we have a large number of harmonics, however, the decrease in the mass increases the “floodness” factor (discussed in detail in revised manuscript): in excessively flooded cells the reversible capacity significantly deteriorates in a short time.

In another comment, the reviewer had expressed concern whether all the material is electrochemically active in our EQCM-D measurements. Fig. S8a and b show typical high initial practical specific capacities in LiTFSI solution. When the floodness factor is high, the initial capacity can also be high but it rapidly deteriorates with the time of cycling whereas in coin cells this high capacity remains much more stable during the entire cycling life.

We made additional measurements with LTO covered on glassy carbon electrode (1 mm diameter) in 3 electrolyte solutions: the floodness factor increased to 5.7×10^5 compared to 1.1×10^4 for the experimental EQCM-D cell with the electrode mass density $88 \mu\text{g}/\text{cm}^2$ (see new Fig. S8).

The most important idea of our work is that independent of the floodness factor the different electrolyte solutions show clearly which electrolyte should be best of all selected, in order to keep high capacity retention. This statement is now proved with the additional experiments on the glassy carbon collector with the highest floodness factor.

5 - This is data from CV performed on three different surfaces, one time each. Although the results make sense to anyone who has worked with these materials, how reliable can the quantification be if this was only done once per electrolyte? I find the lack of error bars on this data very troubling. What would happen if the samples were thinner or thicker, or the experiments simply performed on a different day?

Authors' response: Please see our answer to the previous comment.

6 - You describe a rigid and a less rigid layer for the LTO and explain that this difference is based on the distance from the sensor. What are the two thicknesses? In the schematic in Figure one they appear to be equal thicknesses. This should be clarified.

Authors' response: The logic of EQCM experiments and the essence of our answer to the reviewer's comment 4 suggest that the electrode mass density rather than thickness is the relevant parameter. Generally, EQCM is the mass density sensor rather than thickness sensor if density is not constant or is unknown. QCM serves as a very precise thickness sensor for thin film deposition in the vacuum only when these films are uniform and have the theoretical specific density (a sufficient condition for the correct measurement in the air) and, additionally, should be entirely flat for the measurements in liquids (for details – see our recent review (Electrochim. Acta (2017) 232, 271-284).

QCM, therefore, cannot assess the density and thickness of the coating independently. It is not by chance that even in the gravimetric QCM sensing (rather than viscoelastic) the so called Sauerbrey thickness is used instead of the true physical thickness – all this is about the unknown density when the coatings are porous or subjected to uncontrolled swelling. In these cases, a special analysis is required to distinguish between the contributions to the frequency change due to swelling (inertial effect) or swelling accompanied by real softening (change in complex shear modulus).

Viscoelastic measurements and modeling do not change this general rule concerning the density - thickness relationship. The main goal of the viscoelastic analysis of the LTO coatings is to separate the responses due to the acoustically rigid layer from that due to a single / few viscoelastic layers with different shear moduli. If the coating would contain physically 3-4 layers of the intercalations particles but the binder is absolutely rigid, QCM “sees” this multilayered structure as a single acoustically rigid layer. The LTO coating described in the paper consists of one rigid and one viscoelastic layer. It is by chance that the number of acoustic layers in the coating appeared to be close to the number of the physical layers. The method of analysis of EQCM-D characteristics of intercalation electrodes containing layers with different mechanical properties is called acoustic multilayer formalism (described in full detail in the revised manuscript).

7 - I see no addressing of the impact of electrolyte inclusion into the pores of the structure. This is important because the porosity of the electrodes is not reported, and the electrodes are of varying thicknesses. Address how this factor affects your results and conclusions. Keep in mind that the electrolytes have different wetting abilities.

Authors' response: The reviewer is right, and this comment relates to the very essence of our acoustic analysis. In the revised manuscript when interpreting the shift between the experimental values of $\Delta f/n$ in solution with respect to that calculated by the viscoelastic model, attention was paid to the fact that the difference is due to the inertial contributions to $\Delta f/n$: the total frequency shift in Fig. S2b is 5.2 kHz. The major contribution is due to the dry mass of the electrode (5 kHz). Hence the difference 200 Hz can be ascribed to filling of the narrow pores of the bottom rigid layer: only the pores with a width much less than penetration depth provide the inertial contribution to the frequency change (dissipation is not involved).

Note the important difference with the porosity of the top viscoelastic layer: whatever pores are inside this layer, the solution inside them is accounted for by the effective homogeneous viscoelastic model. This contribution cannot be separated because QCM "sees" the viscoelastic layer as effectively homogeneous.

8 - What is the LTO particle size after ball milling?

Authors' response: Around 120 nm. Average thickness was obtained by AFM 420 nm. Hence this is equivalent to 3.5 physical layers and two effective acoustic layers: the bottom rigid and the top viscoelastic layer.

9 - CV does not imply that all of the LTO reacts with lithium (unlike a slow galvanostatic experiment). What is the actual amount of lithiation and delithiation based on the coulometry? This is needed information because the reader wants to know if the "rigid" LTO is even being lithiated/delithiated during the CV.

Authors' response: The samples for EQCM-D measurements are extremely thin particulate coatings easily accessible to penetration of the electrolyte solution. The reviewer's comment about the difference in CV and galvanostatic methods of charging is truly valid for the thick practical electrodes.

The new Fig. S8 reports the practical specific capacities of LTO in 3 electrolyte solutions for EQCM-D samples (158 mAh/g for LiTFSI), and even practical specific capacity was estimated for the LTO coating on a glassy carbon current collector with the maximal floodness factor 5.7×10^5 : the capacity was ca. 131 mAh/g. The excessively flooded cells are able to ensure high initial specific capacities of the electrode but are unable to keep these high values for a long period of time because of the continuous parasitic reactions (having of course finite kinetics). In turn, the occurrence of the excessive parasitic reactions so nicely seen in Fig. S8b for LiPF_6 and $\text{LiPF}_6 + 2\% \text{VC}$ for LTO on glassy carbon is due to a poor quality of the SEI because of the large floodness factor. However, despite

the fact that the quality of SEI does depend on the floodness factor, the potential ability of the electrolyte salts (and solvents, we believe) to form protective SEI is seen at any floodness factor and quantified in a parallel EQCM-D characterization. The difference in the effect of the electrolytes on the quality of SEI is not as obvious from the experiments with practical coin cells: many hundreds of cycles are required to see a significant difference. In contrast, half-an-hour or one hour is sufficient to perform EQCM-D characterization quantifying the mechanical properties of the SEI forming in different electrolyte solution.

10 - References are needed for the sentence in lines 120-121. I have seen surface oxides lithiate at 1.5 V in the literature, and the three electrolytes have different onset SEI formation potentials. References needed.

Authors' response: We have performed a special literature search using WOS, Science Finder and also Google Scholar. We did not find any electrochemical data for LiTFSI in organic carbonate solutions. What we really found that is the paper by Yuxuan Zhu et al. "Enhanced cycle performance of $\text{Li}_4\text{Ti}_5\text{O}_{12}$ anode in ether electrolyte induced by the solid-electrolyte interphase film" RSC Advances, 5, 56908-56912 (2015). The paper describes very good electrochemical performance of LTO (chronoamperometry and electrochemical impedance) in an ether solution of LiTFSI. We did not cite this paper because the authors awkwardly selected the reference system: LTO cycled in LiPF_6 in EC+DMC solution rather than LiPF_6 in the same ether solution (one should, of course, check the compatibility between the both components). Since the authors state that the excellent cycling behavior of LTO in ether solution of LiTFSI relates to the SEI formed by solvent decomposition (ether) this makes the comparison a bit confusing: LiPF_6 (at least in EC+DMC solution as follows from our work) is responsible for the lower quality of the forming SEI. The EQCM-D method is very suitable to comprehensively characterize (quantify) the difference originated from reactivity of different solvents.

Another excellent paper entitled "On-Line Electrochemical Mass Spectrometry Investigations on the Gassing Behavior of $\text{Li}_4\text{Ti}_5\text{O}_{12}$ Electrodes and Its Origins", by R. Bernhard et al. J. Electrochem. Soc., 161 (4) A497-A505 (2014) describes the behavior of LTO using LiTFSI in a mixture of EC+EMC but only after addition of small amounts of water (LiTFSI was used here with the purpose to avoid parasitic reactions caused by PF_6^- in the presence of water).

11 - In line 192, it seems odd to call SEI formation in the presence of VC "impressive". That's a qualifying statement that detracts from the scientific nature of this work. Do you mean to imply that this SEI is the best?

Authors' response: The reviewer is right, we deleted this word. Actually, it related only to the ability of EQCM-D to sense a local change in viscosity in the vicinity of the

electrode apparently as a result of the polymerization reaction. No, this does not mean that the SEI is the best, only the sensing ability was meant as indicated above.

12 - Address if the 150 degree treatment of the electrodes has any negative effects on the binder. It's notable that for the coin cell electrodes you only went to 100 degrees. That's because 150 degrees can be too hot for this binder system.

Authors' response: The difference mentioned by the reviewer is quite natural for the fabrication of thick practical composite electrodes (e.g. starting from 5 mg/cm² mass density) and for the thin particulate composite coatings for EQCM-D study: 0.088 mg/cm². The success of EQCM-D experiment consists of rigid attachment of the electrode coating to quartz crystal surface (rigid attachment is not the same as the bulk stiffness of the coating, the coating can be viscoelastic throughout its bulk but being attached rigidly to the quartz crystal surface. It is the interfacial property (called also “non-slip” condition) rather than the bulk one. According to our experience, 150 ° C ensured rigid attachment during airbrush method of the coating fabrication. A high practical specific capacity close to the theoretical one is evident that PVdF neither experiences degradative mechanical decomposition nor deterioration of the electrical properties of the electrode.

100° C for fabrication of thick electrodes is natural if one does not want to see an immediate mounting up of the active mass detaching it from the current collector. The reviewer is certainly aware of very fine (sometimes tricky) conditions ensuring good adhesion of the active electrode mass to the current collectors for the coin cell measurements. One should take into account that we have to form rigidly attached layers of intercalation particles to the absolutely flat surface of a Cu-coated crystal without application of the external pressure (calendaring). This makes a huge difference compared to the surface of Cu foil for the fabrication of composite electrode by doctor blade method.

13 - How was the water level in the electrolyte determined?

Authors' response: The water content was determined by Karl–Fischer titration (899 Coulometer, Metrohm).

14 - In the EQCM, was Li the counter electrode and reference? If so, how was it adapted into the cell. Was the counter/reference something else, a pseudo? If so, what was it?

Authors' response: Both real Li electrodes, Li CE on Ni mesh.

15 - Line 367, you mean non-aqueous.

Authors' response: Following the reviewer's advice, we have tested a neat PVdF film in aqueous Li_2SO_4 and $\text{LiPF}_6/\text{EC}+\text{DMC}$ solutions: see our answer to your comment 3 with the embedded figure. PVdF is slightly viscoelastic in aprotic solutions. This does not contradict the acoustic concept that we used since we sensed the viscoelastic properties of a thick ($80 \mu\text{g}/\text{cm}^2$) PVdF film whereas the polymeric binder exists in the composite electrode as thin walls (partitions) between the electrode particles and particle-binder interactions may effectively strengthen its mechanical properties. The conclusion about the ability of EQCM-D to distinguish between two acoustic layers, one rigid and the other viscoelastic remains intact. The intrinsic mechanical property of the binder is important (i.e. the experiment with the neat PVdF film) but the properties induced by the particle-binder interactions and especially the local distribution of the binder around the intercalation particles are important as well. HR SEM of the particulate composite electrodes (not only of the LTO but also a few other electrodes that we studied as well) leaves no doubt about the preferentially rigid character of the bottom layer and softer top layer using PVdF binder. Moreover, even thicker LFP electrodes (mass density $> 120 \mu\text{g}/\text{cm}^2$) with PVdF binder tested in aqueous Li_2SO_4 solution reveal viscoelastic top layer: this shows that the local distribution of the polymeric binder is a very important factor of mechanical behavior of thin composite electrodes.

16 - Figure 4 needs a thickness axis. Writing the number in nanometers is not enough. The bars being shorter for the VC case lead to an erroneous first interpretation. Axis needed.

Authors' response: Thank you, corrected.

REVIEWERS' COMMENTS:

Reviewer #1 (Remarks to the Author):

I am convinced that the authors have responded to the points raised by the advisors correctly and that this has improved the manuscript. Therefore, I consider that the manuscript is ready to be published.

Reviewer #2 (Remarks to the Author):

The responses to me were sufficient. However, I only wish that more from the responses to me were incorporated into the final manuscript. For example, I don't see any mention of the materials particle size after ball milling (forgive me if I'm wrong). Readers will be interested.